# Antibiotic Elution from Cement Spacers and Its Influencing Factors

**DOI:** 10.3390/antibiotics14070705

**Published:** 2025-07-14

**Authors:** Bernd Fink, Kevin D. Tetsworth

**Affiliations:** 1Department for Joint Replacement, Rheumatoid and General Orthopaedics, Orthopaedic Clinic Markgröningen, Kurt-Lindemann-Weg 10, 71706 Markgröningen, Germany; 2Orthopaedic Department, University Hospital Hamburg-Eppendorf, Martinistrasse 52, 20251 Hamburg, Germany; 3Department of Orthopaedic Surgery, Royal Brisbane and Women’s Hospital, Level 7 NHB, Herston, QLD 4029, Australia; ktetsworthmd@gmail.com; 4Limb Reconstruction Centre, Macquarie University Hospital, Suite 303, 2 Technology Place, Macquarie Park, Sydney, NSW 2109, Australia; 5School of Medicine, Biruni University, 34000 Istanbul, Turkey

**Keywords:** periprosthetic joint infection, spacer, antibiotic elution, antibiotic-loaded bone cement

## Abstract

Antibiotic-loaded cement spacers play a crucial role in two-stage revision arthroplasty of infected total hip and knee prostheses. There is still controversy regarding whether the elution from antibiotic-loaded cement spacers is greater than the MIC for a prolonged time between stages. Therefore, the aim of the current review was to determine how long spacers elute antibiotics above the MIC for most causative microorganisms, as well as to evaluate what factors influence that elution. Independent of methodological differences and weaknesses of the studies themselves, several study results indicate that after an early peak of antibiotic release from the spacer in the first 1 to 2 days (followed by a gradual decline), a sufficient release above the MIC for most causative bacteria continues for 6 to 12 weeks.

## 1. Introduction

Periprosthetic joint infection (PJI) is a devastating complication after total joint arthroplasty, with a prevalence between 0.85% and 2% for hip and knee replacement [1,2]. It is the third most frequent indication, after aseptic loosening and dislocation, for revision total hip arthroplasty, the main cause of revision total knee replacement, and has an associated mortality rate of 8–25% per year [3,4,5]. Most prosthetic joint infections are caused by aerobic Gram-positive bacteria, with staphylococci and streptococci isolated in 65% to 85% of cases, followed by aerobic Gram-negative organisms in 6% to 23%, and anaerobic bacteria in approximately 12% [6,7]. Culture negative PJIs represent between 6 and 12% of cases, and fungal infections are identified in only 2% [6,7]. In late periprosthetic joint infection (later than 4–6 weeks after implantation), all foreign material must be removed because of the formation of biofilms on the implants by the responsible microorganisms. Two-stage revision surgery has been a mainstay of treatment for the past several decades, and an interim phase of several weeks (typically 6–12 weeks) with or without a spacer remains the most frequently used strategy for management of late PJI [1,4,5].

The benefits and goals of using spacers in septic two-stage revision arthroplasty include the maintenance of joint mobility and limb length, the limitation of scar formation and soft tissue contraction, which make reimplantation surgery at the second stage easier, and local antibiotic delivery directly to the source of infection [8,9]. Spacers used can be commercially available, prefabricated and preloaded with antibiotics (a single antibiotic most often with gentamicin or vancomycin), or they can be constructed intraoperatively with or without custom-made molds (commercially available or not). With respect to the hip, they can act either as a hemi-arthroplasty or instead as two-part articulating spacers with both a spacer cup and a spacer stem [8,9]. Regarding the knee, they can be either static or articulating as a two-part spacer, and some surgeons have also combined a re-sterilized femoral component with a tibial cement spacer [8,9,10,11,12,13,14,15].

The advantage of antibiotic-loaded cement spacers (ALCSs) is that the concentration of the antibiotic at the site of infection is much higher (up to 700 times) than that achieved by systemic administration [3,9,16]. At the same time, complications that arise from high systemic doses are avoided [3,9]. Moreover, the antibiotics provided by the spacer diffuse locally into tissues with decreased blood supply, where it would otherwise be impossible to reach the Minimum Inhibitory Concentration (MIC) for the causative microorganism through systemic administration [3,9]. 

The local antibiotic concentrations around the spacer should be at least higher than the MIC for that antibiotic against the causative microorganism(s) [9,17,18,19]. These concentrations should be maintained at this high level for at least the time between the two stages (until spacer removal) in order to eradicate the microorganisms that may remain locally after the debridement and removal of the infected foreign materials during the first stage, and to minimize the risk for infection recurrence or emergence of resistant strains [17,19,20]. The risk of developing microbial resistance and re-colonization of bacteria on the spacer may increase when the concentration of the antibiotic drops below therapeutic levels [21,22,23,24,25].

There is still controversy regarding whether antibiotic elution from ALCS is greater than the MIC for a prolonged time between stages, or whether ALCS could generate resistance and lead to secondary superinfection by these so-called “bacterial resisters”. Should any bacteria have survived after the first stage (bacterial resisters), bacterial growth on the spacer surface might be possible. Some in vitro and animal studies have identified the ability of bacteria to adhere to and/or colonize both plain and antibiotic-loaded bone cement [22,25,26,27,28]. These bacteria could be a source of recurrent infection and may be responsible for failure of the second-stage re-implantation [29,30]. Therefore, the aim of the current review was to determine how long antibiotics elute from spacers above the MIC for most causative microorganisms, as well as to evaluate what factors influence that elution.


**Factors influencing the elution of antibiotics from methacrylate spacers**


Elution of the antibiotic from a methacrylate spacer takes place through a diffusion mechanism. Initially, the antibiotic is eluted from the surface, but its later release occurs from the internal portion of the spacer via a network of cracks, channels, and cavities that are formed by forces created by compression and friction during weight bearing [31,32,33]. Most authors observe two phases in the elution process of the antibiotic. The first phase, described as the burst release phase, occurs minutes to hours after spacer implantation, and is characterized by very high local antibiotic release with its peak after 24 h [16,21,32]. This rapid release is based on the elution of the antibiotic from the surface of the spacer and it depends on the surface roughness of the bone cement. The second phase, following the initial burst release, is characterized by a significantly lower antibiotic concentration, that is maintained for a prolonged period of time. This is considered the sustained release phase and is the result of penetration of water into the hydrophilic polymethylmethacrylate (PMMA), “washing” the water-soluble antibiotics out of the spacer [21]. This process is dependent on the bulk porosity of the cement [34]. The bolus of antibiotics in the first phase is immediately delivered to eradicate any bacteria, presumably followed by sustained drug release well above the MIC to kill any remaining bacteria [35]. Therefore, antibiotic release from bone cement depends on both the surface and the water-absorbing properties of the cement [9,36]. Others have described three phases: the exponential phase (during the first few days postoperative), the declining (middle) phase, followed by the final low constant-elution phase [31]. The antibiotic concentration during the exponential phase is most dependent on the antibiotic dose that is loaded in the spacer. The potential of the tissues around the spacer to absorb the antibiotic and any joint motion may also affect the quantity of the antibiotic released from the spacer [9,32,33]. Several factors affect the release of antibiotics from bone cement such as the type of methacrylate, the quantity and the ratio of the incorporated antibiotics, as well as the porosity of cement, the surface characteristics and geometry of the cement body, the way the cement is prepared, and the environmental circumstances [34,37]. These factors all need to be taken into consideration to determine if ALCS elute antibiotics at a level greater than the MIC for causative microorganisms for the entire time the spacer remains in situ.


**Antibiotics and antibiotic combinations for impregnation of ALCS**


Not every antibiotic can be incorporated into a spacer. The antibiotics should be heat resistant (up to 82–83 °C for 12–13 min), so that they are not deactivated during the PMMA polymerization process [9,38,39]. They should be water soluble, to be “washed out” and dispersed into the tissue surrounding the spacer [9,37], and antibiotics with a lower molecular weight are typically more water soluble [17]. The antibiotics used should be available in powder form because liquid antibiotics (even though cheaper and leading to potentially higher elution than powder) reduce the mechanical stability of the bone cement significantly [9,20,40]. For example, adding liquid gentamicin to the cement reduces the compression strength by 49% and the tension strength by 46% [40]. Antibiotics should be chemically stable (neutral) and should not react with other molecules contained inside the cement. They should have a broad antibacterial spectrum and should possess bactericidal properties, even in low concentrations. The antibiotics should have a low MIC for the targeted microbes. Moreover, they should have no risk, or at least a low risk, of allergic reactions or delayed hypersensitivity, should have a low serum protein binding, and not promote the formation of resistant microorganisms [17,21]. Finally, the antibiotic should have little or no influence on the mechanical properties of the cement [37]. 

Aminoglycosides (gentamicin and tobramycin) and glycopeptide antibiotics (vancomycin and teicoplanin) are most appropriate for incorporation into cement, and are effective against a broad spectrum of microorganism commonly isolated in PJI [9,17]. Except for these commonly used antibiotics (gentamycin, tobramycin, vancomycin), antibiotics appropriate for ALCS also include amikacin, amphotericin B, fluconazole, cefazolin, cefotaxime, cefuroxime, ciprofloxacin, clindamycin, colistin, daptomycin, erythromycin, linezolid, meropenem, piperacillin/tazobactam, teicoplanin/tazobactam, and ticarcillin. These antibiotics are all reasonable options when the causative microorganisms have been identified preoperatively [31,38,41]. Rifampicin is a very effective antibiotic against biofilm produced by staphylococci. However, ALCS with rifampicin is not recommended because rifampicin acts as a radical scavenger and therefore inhibits the polymerization process, altering the cement consistency during polymerization and resulting in decreased mechanical strength [9,42,43]. On the other hand, tetracycline, while effective against both Gram-positive and Gram-negative bacteria, is heat inactivated and may instead promote bacterial resistance [21,41,44].

The elution of different types of antibiotics from the cement is variable. Wahlig found that gentamicin possessed the best release characteristics from bone cement on the basis of elution concentration level and duration [45]. Antibiotics are either eluted with a burst or with a continuous pattern. Vancomycin is a typical example of an antibiotic with burst release kinetics, with a high initial release followed by a steep decline. In contrast, gentamicin is typically released continuously [46]. However, the transition from one release kinetic to the other is smooth. Galvez-Lopez et al. [44] demonstrated, by comparing the elution kinetics of 11 different antibiotics, that each antibiotic exhibits its own individual release behavior. For example, meropenem exhibits a continuously declining release over a long period of time, while gentamicin is continuously released with almost no decline. Moreover, moxifloxacin is notable for a longer burst release than vancomycin [47]. One explanation for this is the differences in the molecular weights of the antibiotics; for example, 467 g/mol for tobramycin and 1449 g/mol for vancomycin. A higher molecular weight appears to inhibit the mobility of an antibiotic through the polymer matrix, a feature that would influence the elution properties [48]. Besides a high molecular weight, other factors may be responsible for the lower elution rate of vancomycin and ofloxacin in comparison to, for example, gentamicin and clindamycin. These factors include the physicochemical characteristics of the antibiotics, such as charged, polar, or hydrophobic side chains; interference with cement polymers; and the stability of the drug in the presence of polymerizing heat and biological fluid, as well as the different consistency of the cement itself (degree of porosity, roughness, size of preparation, and surface area) [49]. 

Adding more than one antibiotic to the cement has many advantages over monotherapy. Apart from broadening the antimicrobial spectrum, synergistic effects have been described that lead to an overall increased release of each antibiotic from PMMA bone cement, resulting in a stronger antimicrobial effect [9]. With the use of dual-loaded antibiotic bone cement, the second antibiotic acts as a soluble filler, resulting in increased PMMA porosity. The magnitude of this synergistic effect is dependent on the relative loading ratios of the antibiotics [37]. This synergistic phenomenon, named passive opportunism, has been proven, for example, for the combinations of gentamicin and clindamycin [50,51], for gentamicin and vancomycin [19], for gentamicin and fucidic acid [51], for tobramycin and vancomycin [48,52,53,54,55], for aztreonam and vancomycin [16], for gentamicin and linezolid [56], for meropenem and vancomycin [57,58], and for daptomycin and tobramycin [59]. For example, Hsieh et al. [20] analyzed the elution of gentamicin and vancomycin from Simplex^®^ bone cement. The combination of both antibiotics increased the release of gentamicin by 45%, and of vancomycin by 145%, respectively [20]. Penner et al. [52] demonstrated that the combination of tobramycin and vancomycin in PMMA enhances the release of tobramycin by 68% and of vancomycin by 103% in comparison to controls containing tobramycin or vancomycin alone. Paz et al. [60] investigated the interaction when adding more than two antibiotics to the cement. The addition of cefazolin significantly increased vancomycin elution from an ALBC containing gentamicin and vancomycin [60]. However, release kinetics depend not only on the combination of antibiotics, but also on the relative masses of the combined antibiotics in the cement. For example, gentamicin release significantly increases when the proportion of vancomycin in the cement is raised [36]. However, it must be pointed out that the interactions of antibiotic release are dependent upon many factors, such as the cement brand as well as the antibiotic combinations and the proportion in the cement [55,61]. Boelch et al. investigated antibiotic elution from vancomycin and gentamicin-loaded Palacos R+G (Heraeus Medical, Wehrheim, Germany) and Copal Spacem cement (Heraeus Medical, Wehrheim, Germany). For both cements, vancomycin elution was significantly increased when the amount of vancomycin in the cement powder was increased [61].

Increasing antibiotic concentrations results in greater antibiotic elution not only by a simple increase in the concentration gradient for diffusion, but also by an increase in the surface area of the cement resulting from greater porosity [62]. However, antibiotic loading also reduces the mechanical integrity because antibiotic molecules interfere with the curing of the cement. As already mentioned, liquid antibiotics in particular interfere extensively with the polymerization process, and as a result reduce the mechanical stability of the cement significantly. Therefore, the use of antibiotics as a powder is recommended, and generally preferred for clinical use [17,20,63]. The more antibiotic added, the more the structural integrity of the cured cement will be reduced. The amount of reduction also depends on the cement brand and cement type. Some have recommended a maximum antibiotic concentration of 20% weight/volume (*w*/*v*) in spacers because the mechanical weakness of the spacer could be managed by postoperative restricted weight bearing [63]. However, this effect cannot be controlled and the duration the spacer remains in place may not be predictable, or it may even be permanent in those cases where the second revision procedure is not warranted. It is important to recognize that the antibiotic elution itself may further reduce the mechanical strength of ALCS over time, although not necessarily [36]. Therefore, manual antibiotic loading above a cumulative antibiotic proportion of 10% ***w/v*** of the cement powder is not recommended for the preparation of a temporary spacer because it then no longer satisfies the mechanical ISO requirements for bone cement [17,36,64,65,66,67,68,69].

## 2. Porosity of the Cured Cement

Pores in the cement matrix increase the surface area from which antibiotic can elute, and therefore the antibiotic elution rate out of the spacer itself [70]. Mixing the cement monomer and polymer with the additional antibiotics by hand instead of vacuum mixing or centrifugation mixing increases the porosity of the cement by creating air bubbles, and thus increases the elution of the antibiotics from the cement [71]. Moreover, Neut et al. [72] reported that hand mixing with a spatula resulted in much larger total antibiotic release than hand mixing in a special system (Cemvac System). However, the influence of hand or vacuum mixing techniques on antibiotic elution depends on further factors, such as water solubility of the antibiotic, the diffusion gradient, and above all the choice of cement [55,73]. On the other hand, hand mixing leads to more inhomogeneous antibiotic distribution in the cement matrix and can therefore lead to inconsistent antibiotic elution [74]. McLaren et al. [75] compared different hand-mixing techniques for manual loading with the corresponding premixed cement formulation, but could not demonstrate that hand mixing produced a consistently “dissimilar homogeneity of antibiotic distribution”.

The porosity of the cement can be modified by adjusting the extent of homogenization of the antibiotic and the powder. Miller et al. [76] produced a highly porous ALBC through the addition of vancomycin chunks, resulting in significantly greater antibiotic release. Special additives to the cement can also modify the porosity of the cement further [77]. For example, adding gelatin results in pore formation in PMMA cement [78,79]. Furthermore, calcium phosphate can increase the porosity of bone cements, thereby enhancing antibiotic release [80]. However, to date, these porogens are not in routine clinical use. In contrast, calcium carbonate, a biodegradable and soluble substance, is in clinical use as a component of the commercially available bone cement Copal Spacem (Heraeus Medical, Wehrheim, Germany), which is particularly intended for use as a spacer. A higher microporosity of Copal Spacem cement is reported in contrast to the Palacos R+G cement (Heraeus Medical, Wehrheim, Germany), resulting in better elution of several antibiotics [49]. However, this could not be confirmed when a combination of vancomycin and gentamicin was loaded into these two cements [61]. In conclusion, an easy way to modulate porosity in the operating theater would be to adjust the homogenization of the antibiotic and the cement.

## 3. Choice of Cement

Bone cements differ in their compositions depending not only on the intended use but also on the manufacturer (Table 1). Commercially available ALBCs have well-defined proportions of the incorporated antibiotic, are available with antibiotic combinations, and have specific antibiotic elution properties, because cements of different formulations usually possess different degrees of cross-linking and homogeneity that can affect the elution of the incorporated drug [55]. As a result, each cement brand and type has its own release behavior. For example, Palacos cement (Heraeus Medical, Wehrheim, Germany) exhibits higher and longer antibiotic elution of tobramycin and vancomycin in comparison to Simplex cement [55,62]. The comparison of medium viscosity Palacos R+G and high viscosity Palacos R+G is typical of cement specific release: although both cements contain the same proportion of antibiotic, more gentamicin is released from medium viscosity Palacos R+G [81].

## 4. Geometry of the Cement Body

Antibiotic release from PMMA is a surface phenomenon and does not depend on the amount of the cement [82], but instead the geometry of the implanted cement body does influence the antibiotic elution rate. The larger the cement surface in contact with the surrounding tissues, the greater the rate and amount of antibiotic elution from the spacer [21,31,82,83]. Duey et al. [84] compared different cement geometries and volumes for the release of vancomycin and tobramycin from SimplexP bone cement bodies and reported a positive linear correlation of the elution with the surface area. On the other hand, antibiotic release did not correlate with specimen volume [84]. These findings were confirmed by the results of Masri et al. [85] during an in vitro study on Simplex P bone cement. They demonstrated a significant increase in tobramycin elution when the surface area was enlarged but the cement volume remained constant. In contrast, when the cement volume was reduced but the surface area remained the same, they did not observe a significant change in the elution pattern [85]. This phenomenon is explained by the fact that antibiotic elution happens predominantly from the outer layers of the cement. Almost the entire antibiotic is eluted from the superficial 100 micrometer layer, while only 19% of incorporated antibiotic is released from the deeper 700 micrometer layer [82]. These observations explain why beads have higher antibiotic elution compared to structural spacers of the same composition, because beads have a greater surface area to volume ratio, with surface elution from PMMA responsible for initial burst levels [43,86]. Bearing this in mind, spacers should ideally be formed with the largest possible surface area, while maintaining mechanical integrity.

## 5. Cement Mixing Technique

The mixing procedure of the antibiotic and the cement also affects antibiotic elution from the spacer. As already mentioned, hand mixing under normal atmospheric conditions leads to higher porosity, lower spacer strength, and an increase in the outer (antibiotic-releasing) surface of the spacer. Due to the higher porosity and the larger external surface, the elution of antibiotics from the spacer is increased. Hand mixing, in comparison to vacuum mixing, can increase peak antibiotic concentration up to five-fold in vitro [72,87]. On the other hand, with hand-made spacers, the antibiotic may not be released uniformly. In contrast to hand-made spacers, the preparation of commercial spacers is standardized and the ingredients are mixed far more homogenously [31]. Vacuum mixing increases the mechanical strength of the cement by decreasing the cement porosity and, theoretically, the release of the antibiotic. On the other hand, other factors, such as water solubility of the antibiotic or local osmosis, play a further role in the elution characteristics and may increase antibiotic elution from vacuum-mixed cement [88].

Two different and, in part, contradictory recommendations for adding the antibiotic to the cement powder have been described; the first technique includes a thorough homogenization of the antibiotic powder and the cement powder. The crystalline antibiotic is ground in a mortar and then added to the cement powder in several steps. This mixing technique is designed to provide consistent and reproducible cement properties after the cement cures [64]. In contrast, the other technique describes adding the antibiotic to the cement powder all at once, resulting in only rough homogenization. This concept is thought to provide maximum antibiotic elution after hardening due to the formation of antibiotic clumps in the cement [89]. Laine et al. [90] compared the effects of different cement mixing techniques that give varying levels of homogenization, and dispensing with the homogenization process almost entirely induced greater pore formation. Nevertheless, subsequent mechanical testing failed to demonstrate any significant difference between the mixing techniques.


**Time of sufficient elution from spacers**


All these factors influencing the relationship between antibiotics and bone cement have to be taken into consideration when interpreting the results of the many in vitro and in vivo studies of the elution characteristics of antibiotics from spacers. Because there are great differences in the methodological details, these studies cannot easily be compared and sometimes the different results simply reflect these methodological differences. To try to determine how long antibiotics elute from spacers at a therapeutic level, a wide range of selected studies were evaluated. Reports were chosen that provide results of elution characteristics where antibiotics were used in powder form, at a cumulative antibiotic proportion of not more than 10%, and using one or more antibiotics in the cement without any other additives (such as glycine, calcium phosphate, or carbonate).


**In vitro studies**


The release of antibiotics from cement spacers has been extensively studied in vitro. In vitro analyses typically involve measurements of antibiotic levels in fluid using predominantly fluorescence polarization immunoassays, or by examination of the antibiotic activity by using zones of inhibition. The majority of these studies have investigated cement devices other than spacers, such as disks [52,91,92,93]. Because the release of antibiotics from bone cement is a surface-dependent phenomenon [94], it is valid to question the relevance of these in vitro observations for hip and knee spacers that have a very different surface geometry. Moreover, the amount and frequency of fluid exchange around bone cement in vitro are variable in these studies. Most protocols change the eluate daily, although some only change this fluid weekly [95]. Greater amounts of fluid and more frequent exchange would influence the elution characteristics observed by changing the concentration gradient [62]. This permanently existing concentration gradient between spacer surface and culture medium does not fully represent the vascular supply, nor the resulting antibiotic diffusion to tissue. Furthermore, antibiotic and physiological differences such as local blood flow and tissue pH may have an influence on the pharmacokinetic activity of spacers, resulting in different antibiotic elution profiles. Moreover, the spacers are tested in vitro in an environment optimized for bacterial growth. Tryptic soy broth is an ideal growth medium and often a new aliquot of the microorganism has been added on a daily basis, whereas the antibiotic amount has not been changed [96]. Therefore, the results of in vitro studies cannot be directly translated into clinical practice and in vivo conditions.

However, nearly all in vitro studies report a very high peak of antibiotic concentration and elution after 24 h, followed by a continuous decline in concentration [47,56,96,97,98,99,100,101,102,103] (Table 2). One of the greatest limitations of several in vitro studies designed to answer the question posed by this review is the duration of the analysis period. Most studies analyze the elution for only a few days and up to one week [56,57,98] (Table 1). For these short analysis periods, sufficient releases of antibiotics over the Minimal Inhibition Concentration (MIC) have been reported [56,57,98]. Boelch et al. [61] observed sufficient release of vancomycin and gentamycin for the test period of 28 days, Galvez-Lopez et al. [47] of different antibiotics for the test duration of 30 days, Oungeun et al. [103] for vancomycin and erythromycin for a period of 42 days, and Kummer et al. [95] for the release of vancomycin, gentamicin, and daptomycin for the test duration also of 42 days.

It may be possible that the spacers behave differently in vivo. The infection localization, the blood perfusion of the infected area, and the protein binding of each antibiotic used are some of the parameters that may actively influence spacer performance in vivo.

**Table 2 antibiotics-14-00705-t002:** In vitro studies.

Author	Reference	Antibiotic	Dosages	Cement	Method	Test Duration	Duration of Sufficient Antibiotic Release
Gálvez-López	[47]	11 different	10% and 20% (4 g and 8 g per 40 g cement); Results for 10%	Medium viscosity cement from DePuy	Beads: HPLC + DAA	30 days	30 days Vancomycin, Gentamicin, Moxifloxacin, Daptomycin, Ertapenem, Meropenem, Cefotaxime
Bitsch	[49]	Ofloxacin, Vancomycin, Clindamycin, Gentamicin	2 g, 4 g and 6 g in 40 g cement	Copal spacem (no antibiotics)	area of clearance	50 days	2, 4 and 6 g for 50 days, Vancomycin: 2 g 20 days, 4 g 43 days
Stevens	[55]	Vancomycin, Tobramycin	different (suff: 3.6 g Tobramycin and 3.0 g Vancomycin = 14.2 wt%)	Simplex and Palacos	DAA with ZI	80 days	14.2 wt%: 80 days MIC; 25 days Tobramycin, 4 days Vancomycin in Palacos
Anagnostakos	[56]	Linezolid		Palacos R+G (0.5 g Gentamicin)	HPLC, FPIA + Photometric (bacterial growth)	8 days	8 days
Andollina	[57]	Vancomycin, Meropenem	different	Cemex XL	CA + microbiological investigation	5 weeks	5 weeks for 1 g Vancomycin + 1 g Meropenem
Boelch	[61]	Vancomycin		Copal spacem (no antibiotics) and Palacos R+G (0.5 Gentamicin)	HEIA	28 days	28 days Gentamicin, Vancomycin
Greene	[62]	Tobramycin, Vancomycin	each 4 g on 40 g Simplex or Palacos R	Simplex or Palacos R	ZI	100 days	100 days Tobramycin, 32 Vancomycin in Palacos
Shiramizu	[70]	Cefazolin	2 g Cefazolin in 40 g cement	Simplex and CMW 3	HPLC	7 days	7 days
Kummer	[95]	Vancomycin, Daptomycin, Fosfomycin	2 g Vancomycin, 1.5 g Daptomycin, 1.5 g Fosfomycin per 40 g cement	Palacos R+G (0.5 g Gentamicin)	LCMS	6 weeks	6 weeks for Vancomycin, Gentamicin, Daptomycin
Anagnostakos	[96]	Gentamicin (G), Vancomycin (V) Teicoplanin	different (0.5 g G + 1 g V in 40 g cement)	Palacos R+G (0.5 g Gentamicin)	FPIA + Photometric (bacterial growth)	20 days	20 days Gentamicin + Vancomycin, 4 days Teicoplanin
Humez	[97]	Daptomycin	1.5 g in 40 g cement	Palacos R+G (0.5 g Gentamicin)	HPLC	2 days	2 days
Moore	[104]	Vancomycin, Tobramycin	2.0 g in 40 g cement	Simplex	LCMS	2 days	48 h = 2 days
Liawrun-rueang	[105]	Gentamicin	2.0 g in 40 g cement	Palacos R+G (0.5 g Gentamicin)	EMIA; FPIA	6 weeks	6 weeks
Allen	[98]	Gentamicin	0.5 g in 40 g cement	Palacos R+G (0.5 g Gentamicin)	DDA	12 days	12 days sufficient
Haseeb	[99]	Ceftaroline, Vancomycin	Ceftraroline 3 wt% (1.2 g), Vancomycin 2.5 wt% (1 g) liquid	SmartSet	HPLC + ZI	8 weeks	6 weeks Ceftaroline, 3 weeks Vancomycin
Ikeda	[100]	Vancomycin	5 wt%	Cemex RX	HPLC	84 days (12 weeks)	56 days (8 weeks)
Slane	[101]	Vanocmycin, Tobramycin	Different	Palacos R	HPLC	28 days	28 days
Goltzer	[102]	Gentamicin	Different	different prefabricated spacers + hand mixed Palacos	NA	7 days	7 days
Salih	[106]	Gentamicin	0.5 g; 1.5 g; 23 g in 40 g cement	Palacos and Copal G+C	LCMS	9 weeks	9 weeks
Oungeun	[103]	Hydrophilic Vancomycin, Erythromycin	55 mg Vancomycin/55 mg Erythromycin	Palacos R	UV-S	6 weeks	6 weeks for Vancomycin and Erythromycin

HPLC = High-Performance Liquid Chromatography; LCMS = Liquid Chromatography Mass Spectrometry; EMIA = Enzyme Multiplied Immunoassay; FPIA = Fluorescence Polarization Immunoassay; ZI = Zone of inhibition (bacterial growth); HEIA = High Enzyme Immunoassay; DDA = Disc Diffusion Assay; NA = Ninhydrin Assay; CA = Chromatographic Assay; UV-S = Ultraviolet visible Spectrometry; g = gram; wt% = weight percent.


**In vivo studies**


Due to the limitations of the in vitro studies, the more important studies to determine if there is a sufficient elution period from spacers are the in vivo studies. Here, methodologically, this research can be separated into animal in vivo models and human in vivo investigations. The latter can be further distinguished between those studies where antibiotic concentrations were measured in joint fluid drains for a few days after surgery, and those where the spacers removed during the second surgery of the two-stage procedure are analyzed for remaining antibiotic elution capacity.

Based on the discussion above, it is apparent that a number of important factors have an influence on the release kinetics and elution period from bone cement in vivo [9]. These factors have to be taken into consideration for the interpretation of the elution of antibiotics from ALCS. Generally, the type and porosity of cement, the preparation of the cement, the type and ratio of antibiotics, the quantity of antibiotic(s), addition of one or combination of two or more antibiotics, the method of mixing the antibiotics in the cement, the roughness and total surface area of the spacer, the duration of spacer implantation, spacer geometry and surface configuration, spacer articulation, and the environmental circumstances specific to the clinical setting are all recognized factors with a possible effect on the antibiotic release from bone cement [9,34,37,85]. Moreover, articulating spacers lead to abrasion of cement particles [107], and this abrasion leads to new surfaces on the spacer from which antibiotics can elute, which may be associated with increased and prolonged antibiotic release [49,108]. Furthermore, an increase in the distance between the cement and the surrounding tissues results in decreased concentration of the antibiotic in the tissue [83]. The type of tissues around the spacer also affects antibiotic elution from the spacer and antibiotic diffusion into the adjacent inflamed tissues. Cancellous bone absorbs more antibiotic than cortical bone, while a hematoma (dead space) that potentially lies between the spacer and the bone or soft tissues absorbs more antibiotic than cancellous bone [21,83].

Of in vivo studies on animals, those evaluating selected shapes of PMMA spacers particularly warrant mention here. In dogs, Adams et al. [109] noted detectable concentrations of tobramycin and vancomycin from cement beads over a period of 28 days. Similarly, Chapman and Hadley [110] observed a peak of antibiotic release on the first day, but no elution in serum or soft tissue from antibiotic-loaded cement pellets implanted in the intramedullary canal of rabbit femora and quadriceps muscle after 37 days. On the other hand, Bunetel et al. [111] reported sufficient release of gentamycin up to 18 months after femoral implantation of gentamicin (0.6 and 1.2 g per 40 g PMMA) in sheep, while Chohfi et al. [112] noted sufficient release of vancomycin (3 g in 40 g PMMA)-impregnated cement in sheep femora for 3 months. Gatin et al. [113] observed sufficient release of colistin from PMMA following implantation of a silicone elastomer knee implant in rabbits for the full test duration of only 21 days. In contrast to other animal studies, the rabbits in the study by Gatin et al. [113] were exposed to microbial contamination of the implant. The differences in the results reported may simply reflect subtle methodological differences and inherent weaknesses in the design of these studies.

The research models with the most clinical relevance are the clinical studies in humans with spacers in situ. However, an in vivo analysis of the drainage fluid concentration is limited by the time period for which the drainage tubes could be left in place, usually only a few days. Anagnostakos et al. [86] observed sufficient antibiotic elution in the drainage fluid for 7 days when mixing vancomycin in Palacos R+G-cement. As with the in vitro and animal in vivo studies, analysis of the antibiotic concentration in the joint fluid drainage detected a maximal peak after 24 h, well above the MIC of the microorganism for all tested antibiotics, followed by a much lower concentration over the ensuing days [16,86,112,114]. These antibiotic concentrations were mostly above the MIC at the end of the test period (Table 3). However, these concentrations of antibiotics collected from the drains are not fully representative of the pharmacokinetic properties from the spacers, but instead almost exclusively for the intra-articular aspects of the spacer. Particularly with hip spacers, this accounts only for the pharmacokinetic properties of either the head of the spacer alone, or the spacer head in combination with a spacer cup. The antibiotic release from the spacer stem in the femoral cavity cannot be assayed reliably from the joint fluid. Therefore, the measured concentrations represent only one component of the true antibiotic elution in vivo [32]. Fink et al. [108] measured the antibiotic concentration in the spacer membrane around hip spacers 6 weeks after implantation, and noted antibiotic levels above the MIC for gentamicin, clindamycin, and vancomycin. In addition, Bertazzoni Minelli et al. [115] analyzed the antibiotic concentration in the periprosthetic tissue in two cases after explantation of hip spacers and reported concentrations of 12.0 ± 1.5 µg/g for gentamycin and 10.3 ± 1.5 µg/g for vancomycin, values higher than those determined by the elution studies. These antibiotic concentrations obtained from tissue specimens at the infection site exceed the MICs of most of the common pathogens involved in prosthetic joint infections, suggesting antibiotic elution clinically is more than adequate to serve its purpose.

Generally, because elution for longer time periods in vivo could not be measured through analysis of drainage fluid, the analysis of spacers after removal at the second stage of septic two-stage revision arthroplasty would seem to be a more appropriate means to determine if antibiotic elution exceeds the MIC for a prolonged period. In fact, joint aspiration at the time of spacer removal and analysis of the antibiotic concentration in the joint fluid has consistently demonstrated a sufficient concentration above the MIC for 3 months and above in most studies [53,116,117] (Table 3).

The potential for long-term antibiotic elution from spacers can be verified by studies that have evaluated the residual antimicrobial and pharmacokinetic properties of spacers after their explantation in vitro. Bertazzoni Minelli et al. [115] and Kelm et al. [118] have both demonstrated sufficient elution kinetics and bacterial growth inhibition for all tested spacers, independent of their particular implantation period.

One additional feature of hip spacers (hemispacer versus articulating two-part spacers) may influence the antibiotic release during clinical use. Hsieh et al. [117] noted sufficient antibiotic concentrations above the MIC in the joint fluid of 46 hip spacers containing vancomycin and aztreonam after a mean period of 107 days. These spacers were articulating two-part spacers with a cemented stem/cobalt-chrome head articulating on a cement cup. These observations are in concordance with the findings of Fink et al. [108], who reported sufficient concentrations of gentamycin, clindamycin, and vancomycin in the tissue surrounding the spacer after 6 weeks. The spacers used in the latter study were also two-part articulating spacers with a metal head articulating on cement cups. These articulations create wear, as was identified and confirmed by Fink et al. [107] in histological analyses, by X-ray fluorescence spectroscopy, as well as by plasma mass spectrometry. The wear resulted in and created new surfaces, from which additional antibiotic elution can occur. However, Masri et al. [53] also observed sufficient antibiotic concentrations 4 months postoperatively when using Prostalac spacers with tobramycin. Therefore, two-part articulating spacers in the hip (and, by extension, articulating knee spacers) may have advantages regarding antibiotic elution in vivo, and would have the further benefit of preventing acetabular bone resorption and spacer migration (sometimes observed when hemi-spacers are used) [8].

Finally, Griffin et al. [119] analyzed six spacers with different dosages of vancomycin and tobramycin after removal between 37 and 175 days (average 85 days). They detected no biofilm on the spacers using scanning electron microscopy, which may be an indirect sign of sufficient release of antibiotics during these spacer periods; five of these six patients had no reinfection. 

However, one weakness of these in vivo studies is that usually only the most commonly used antibiotics (gentamicin, vancomycin and clindamycin) were tested. Therefore, the release properties of these antibiotics cannot be used to draw conclusions about others. Moreover, the success rate with clinical infection control in these studies was often not disclosed (Table 3).

**Table 3 antibiotics-14-00705-t003:** In vivo studies on humans.

Author	Reference	Antibiotic	Dosages	Cement	Spacer	Method	Test Duration	Duration of Sufficient Release	Infection Control
Hsieh	[16]	Vancomycin + Aztreonam	4 g in 40 g PMMA	Simplex	Two-part hip	Drainage for 7 days, in joint fluid at spacer removal, HPLC	32–156 days, Ø107 days	32–156 days, Ø107 days	97.8%
Masri	[53]	Tobramycin + Vancomycin	3.6 g Tobramycin + 1 g Vancomycin in 40 g Palacos	Simplex P	Two-part hip Two-part knee PROSTALAC	Aspirate joint fluid at spacer removal, FPIA	42–340 days, Ø 118 days	4 months Tobramycin	n.a.
Anagnostakos	[86]	Gentamicin + Vanco	0,5 Gento + 2 g Vanco in 40 g (80 g spacers in 17 patients)	Refobacin/Palacos	Two-part hip	Drainage, FPIA	7 days max	7 days (study period)	n.a.
Fink	[108]	Gentamicin + Clindamycin, Vancomycin	2 g Vancomycin in 40 g Copal-Cement	Copal G+C	Two-part hip	Spacer membrane: (LC-MS/MS)	6 weeks	6 weeks	n.a.
Chohfi	[112]	Vancomycin	3 g in 40 g PMMA (Cerafix)	Cerafix (low viscosity)	THA	Drainage 2–5 days, IEA	5 days max	4 days	100%
Regis	[114]	Gentamicin + Vanco	2.5% Gentamicin, 2.5% (1 g) Vancomycin manually added	Cemex	Hemi-spacer hip (Spacer-G, Tecres)	Drainage, FPIA	1 day (24 h)	1 day (24 h)	n.a.
Isiklar	[120]	Vancomycin	2 g Vancomycin in 40 g cement	n.a.	Hemi-spacer hip	Drainage	2 days	2 days	100%
Balato	[121]	Gentamicin und Clindamycin	1 g Gentamicin + 1 g Clindamycin in 40 g cement	Refobacin Revision Biomet	Two-part knee Hemi-spacer hip	FPIA	2 days	Hip: 48 h Gentamicin, knee: 12 h in 75%, 36 h in 50%	100%
Mutimer	[116]	Gentamicin	0.76 g Gentamicin in 40 g cement	Spacer K, Cemex (Tecres)	Two-part knee	Aspirate joint fluid at spacer removal	99 (63–274)	99 days median (64–274 days) = 3 months minimum	100%
Kelm	[118]	Gentamicin + Vancomycin	0.5 Gentamicin + 2 g Vancomycin in 40 g cement (80 g spacers in 10 hip spacers)	Refobacin/Palacos	Hemi-spacer hip	Drainage + bacterial growth (photometric) after spacer removed	3–14 weeks	Vancomycin 17 days, Gentamicin 14 days	n.a.
Hsieh	[117]	Liquid Gentamicin, Vancomycin	480 mg/20 mL cement monomer; 3 g Vancomycin in 40 g cement	Simplex	Two-part hip	Aspirate joint fluid at spacer removal, FPIA	Ø 87 days (27–128 days)	87 days (27–128 days)	95.2%
Bertazzoni Minelli	[115]	Gentamicin + Vancomycin in holes	0.76 g Gentamicin in 40 g Cemex, 1 g. Vanomycin in 40 g cement	Cemex RX	Hemi-spacer hip Spacer-G (Tecres)	Spacer removed in Phosphate buffer, FPIA	3–6 months	3–6 months Gentamicin	100% of 17 of 20 with reimplantation

HPLC = High-Performance Liquid Chromatography; LC-MS/MS = High-pressure Liquid Chromatography coupled to Tandem Mass Spectroscopy; FPIA = Fluorescence Polarization Immunoassay; HEIA = High Enzyme Immunoassay; IEA = Immunoenzymatic Assay; THA = Total Hip Arthroplasty (no spacer); Ø = average; h = hours; n.a. = not answered.

## 6. Conclusions

Recognizing the various methodological differences and weaknesses of the published studies, the frequent confounding factors influencing the antibiotic elution level, and the variation in the time period implanted all make it extremely difficult to assess with any accuracy the true amount of antibiotic released from spacers under clinical conditions. However, there are several study results indicating that after a burst peak of antibiotic release from these spacers in the first 1 to 2 days (followed by a gradual decline), a sufficient release above the MIC for most causative bacteria continues for 6 to 12 weeks. To obtain more conclusive evidence and data regarding the duration of sufficient antibiotic release from these spacers above the MIC, more systematic in vivo studies are necessary that account for the most important influencing factors, particularly the type of cement used, the type of antibiotics used, the amount of antibiotics used, and the mixing technique employed.

## Figures and Tables

**Table 1 antibiotics-14-00705-t001:** Components of various PMMA cements: √ = present, G = gentamicin, C= clindamycin, V = vancomycin, T = tobramycin, R = irradiation, EO = ethylene oxide and A = aseptic. (Reproduction with permission of Table 13.1 in Kühn KD [36]).

Cements		CMW^®^ 1 G	SmartSet^®^ GHV	Palacos^®^ R+G	Copal^®^ G+C	Copal^®^ G+V	Refobacin^®^ B.C.R	Antibiotic Simplex^®^ P with T	Tianjin Joint Cement
Powder	PMMA	√						√	√
	MA/MMA		√	√	√	√	√		
	Styrene copolymer							√	
	Styrene butadine copolymer								√
	Antibiotic	G	G	G	G+C	G+V	G	T	
	Benzoyl peroxide	√	√	√	√	√	√	√	√
	Zirconium dicoxide		√	√	√	√	√		
	Barium sulfate	√						√	√
	Sterilized	R	EO	EO	EO	R	EO	R	R
	Coloring agent			√	√	√			
Liquid	MMA	√	√	√	√	√	√	√	√
	N, N Dimethyl- (-toluidine)	√	√	√	√	√	√		
	Sterilized	A	A	A	A	A	A	A	A
	Coloring agent			√	√	√	√		
	Hydroquinone	√	√	√	√	√	√	√	√

## Data Availability

Not applicable.

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
