# Peer review of "Antibiotic Elution from Cement Spacers and Its Influencing Factors"

_antibiotics, 2025, doi:10.3390/antibiotics14070705_

Round 1
Reviewer 1 Report
Comments and Suggestions for Authors
please see my comments in the attachment

The English could be improved to more clearly express the research.
Author Response
The changes are in red in the manuscript
Reviewer 1:
Strengths
- Well-structured methodologically – divided into subtopics in a logical and convenient way.
- Covers the theory comprehensively with a high number of references while comprehensively addressing the factors that affect the levels of antibiotics excreted from the cement.
Weaknesses
- The studies presented in humans refer to a limited number of types of antibiotics (those in common use such as Vanco and Gentamicin) so that it is not possible to understand from the review the effectiveness of additional types in humans- need to elaborate.
Response: Thank you for pointing this out. This has been addressed on lines 539-542:
However, one weakness of these in vivo studies is that usually only the most commonly used antibiotics (gentamicin, vancomycin and clindamycin) were tested. Therefore, the release properties of these antibiotics cannot be used to draw meaningful conclusions about others.
- In the studies conducted in humans, it was written about the effective excretion of antibiotics over time from the spacer, but it was not written whether this was expressed in the eradication of the bacteria and if not, what the reasons could be- need to clarify.
Response: We agree with the reviewer, but unfortunately the published studies on the topic often fail to include this information. This has been addressed with an additional column in Table 3 and in Lines 482 and 487: Moreover, the success rate with clinical infection control in these studies was often not disclosed (Table 3).
- The issue of the cement weakening after the antibiotic is excreted from it – they write that there is a loss of strength in the spacer as the antibiotic is excreted from it, even though there are studies that show otherwise – I think there is room to express this in a more reserved manner.
Response: We have tempered our position in this regard with the following statement in line 230-32:
It is important to recognize that the antibiotic elution itself may further reduce the mechanical strength of ALCS over time, although not necessarily [38].
The English could be improved to more clearly express the research.
Response: Thank you for suggesting further edits. The paper was further reviewed by an English native speaking clinical scientist; additional minor revisions and edits to the manuscript have been completed as necesssary to enhance clarity and improve readability.
Reviewer 2 Report
Comments and Suggestions for Authors
The manuscript is a review covering elution of antibiotics from prostetic implants. It is an important area, because implants are very easily colonized by bacteria and once settled, they are notoriously difficult to treat often leading to reoperation and replacement of the implant leading to longer hospitalization and additional health issues. Incorporation of antibiotics into the cement used for fixating the implants is the current solution, but there are unsolved problems, that needs to be adressed and the review gives a comprehensive overview of different types of cement and releaseprofiles, where the common issue is a burst release, where the bulk amount of antibiotic is released within the first short period increasing the risk for development of resistance due to sub-lethal concentrations of antibiotic in the local environment. A comparison of the different cements based on their chemical and structural composition would have been optimal, but is probably not possible due to IP-issues.
Author Response
Changes are in red in the manuscript
Reviewer 2:
The manuscript is a review covering elution of antibiotics from prostetic implants. It is an important area, because implants are very easily colonized by bacteria and once settled, they are notoriously difficult to treat often leading to reoperation and replacement of the implant leading to longer hospitalization and additional health issues. Incorporation of antibiotics into the cement used for fixating the implants is the current solution, but there are unsolved problems, that needs to be adressed and the review gives a comprehensive overview of different types of cement and releaseprofiles, where the common issue is a burst release, where the bulk amount of antibiotic is released within the first short period increasing the risk for development of resistance due to sub-lethal concentrations of antibiotic in the local environment. A comparison of the different cements based on their chemical and structural composition would have been optimal, but is probably not possible due to IP-issues.
Response: Thank you for devoting the time and energy to conduct this review, and we appreciate your suggestion. We have attempted to address this as much as possible, while still respecting various IP considerations; Table 1 has been added with the components of the various PMMA cements commonly available.
Reviewer 3 Report
Comments and Suggestions for Authors
This is a very well written review of antibiotic elution from cement spacers. As the title suggests, it is well focused. It is thorough and provides meaningful conclusions. It is certain that it will be a useful reference for scientists and clinicians, and it should be published with only minor additions.
Throughout the review, the authors discuss differing antibiotics used in bone cement. Use of some is more common that others, and at one point the authors provide a list of antibiotics that have been explored. These include antibiotics with activity limited, primarily, to Gram-positive bacteria, others with broad-spectrum activity (i.e., activity against both Gram-positive and Gram-negative bacteria) and some antifungals. It would strengthen the review to add a brief discussion of the types of organisms that are the predominant infectious agents with medical implants and use this information as a rationale for selection of specific types of antibiotics used.
The authors discuss in vivo experimentation, and much of this involves measurement of antibiotic concentrations in tissues surrounding an implant. Missing is whether there were microbial challenges administered in these in vivo experiments.
Author Response
Changes are in red in the manuscript
Reviewer 3:
This is a very well written review of antibiotic elution from cement spacers. As the title suggests, it is well focused. It is thorough and provides meaningful conclusions. It is certain that it will be a useful reference for scientists and clinicians, and it should be published with only minor additions.
Throughout the review, the authors discuss differing antibiotics used in bone cement. Use of some is more common that others, and at one point the authors provide a list of antibiotics that have been explored. These include antibiotics with activity limited, primarily, to Gram-positive bacteria, others with broad-spectrum activity (i.e., activity against both Gram-positive and Gram-negative bacteria) and some antifungals. It would strengthen the review to add a brief discussion of the types of organisms that are the predominant infectious agents with medical implants and use this information as a rationale for selection of specific types of antibiotics used.
Response: Thank you for this constructive comment. This has been added in the first paragraph of the introduction (Lines 36 -40).
The authors discuss in vivo experimentation, and much of this involves measurement of antibiotic concentrations in tissues surrounding an implant. Missing is whether there were microbial challenges administered in these in vivo experiments.
Response: Thank you for this insightful observation. This was addressed in line 465-66:
In contrast to other animal studies, the rabbits in the study by Gatin et al [115] were also exposed to microbial contamination of the implant.
Round 2
Reviewer 1 Report
Comments and Suggestions for Authors
All points were addressed well